# Optimal Architectures in a Solvable Model of Deep Networks

**Jonathan Kadmon**
The Racah Institute of Physics and ELSC
The Hebrew University, Israel
jonathan.kadmon@mail.huji.ac.il

**Haim Sompolinsky**
The Racah Institute of Physics and ELSC
The Hebrew University, Israel
and
Center for Brain Science
Harvard University

## Abstract

Deep neural networks have received a considerable attention due to the success of their training for real world machine learning applications. They are also of great interest to the understanding of sensory processing in cortical sensory hierarchies. The purpose of this work is to advance our theoretical understanding of the computational benefits of these architectures. Using a simple model of clustered noisy inputs and a simple learning rule, we provide analytically derived recursion relations describing the propagation of the signals along the deep network. By analysis of these equations, and defining performance measures, we show that these model networks have optimal depths. We further explore the dependence of the optimal architecture on the system parameters.

## 1 Introduction

The use of deep feedforward neural networks in machine learning applications has become widespread and has drawn considerable research attention in the past few years. Novel approaches for training these structures to perform various computation are in constant development. However, there is still a gap between our ability to produce and train deep structures to complete a task and our understanding of the underlying computations. One interesting class of previously proposed models uses a series of sequential of *de-noising autoencoders* (dA) to construct a deep architectures [5, 14]. At it base, the dA receives a noisy version of a pre-learned pattern and retrieves the noiseless representation. Other methods of constructing deep networks by unsupervised methods have been proposed including the use of Restricted Boltzmann Machines (RBMs) [3, 12, 7]. Deep architectures have been of interest also to neuroscience as many biological sensory systems (e.g., vision, audition, olfaction and somatosensation, see e.g. [9, 13]) are organized in hierarchies of multiple processing stages. Despite the impressive recent success in training deep networks, fundamental understanding of the merits and limitations of signal processing in such architectures is still lacking.

A theory of deep network entails two dynamical processes. One is the dynamics of weight matrices during learning. This problem is challenging even for linear architectures and progress has been made recently on this front (see e.g. [11]). The other dynamical process is the propagation of the signal and the information it carries through the nonlinear feedforward stages. In this work we focus on the second challenge, by analyzing the 'signal and noise' neural dynamics in a solvable model of deep networks. We assume a simple clustered structure of inputs where inputs take the form of corrupted versions of a discrete set of cluster centers or 'patterns'. The goal of the multiple processing layer is to reformat the inputs such that the noise is suppressed allowing for a linear readout to perform classification tasks based on the top representations. We assume a simple learning rule for the synaptic matrices, the well known *Pseudo-Inverse* rule [10]. The advantage of this choice, beside its mathematics tractability, is the capacity for storing patterns. In particular, when the input

is noiseless, the propagating signals retain their desired representations with no distortion up to a reasonable capacity limit. In addition, previous studies of this rule showed that these systems have a considerable basins of attractions for pattern completion in a recurrent setting [8]. Here we study this system in a deep feedforward architecture. Using mean field theory we derive recursion relations for the propagation of signal and noise across the network layers, which are exact in the limit of large network sizes. Analyzing this recursion dynamics, we show that for fixed overall number of neurons, there is an optimal depth that minimizes the readout average classification error. We analyze the optimal depth as a function of the system parameters such as load, sparsity, and the overall system size.

## 2 Model of Feedforward Processing of Clustered Inputs

We consider a network model of sensory processing composed of three or more layers of neurons arranged in a feedforward architecture (figure 1). The first layer, composed of $N_0$ neuron is the input or *stimulus* layer. The input layer projects into a sequence of one or more intermediate layers, which we also refer to as *processing* layers. These layers can represent neurons in sensory cortices or cortical-like structures. The simplest case is a single processing layer (figure 1.A). More generally, we consider $L$ processing layers with possibly different widths (figure 1.B). The last layer in the model is the *readout* layer, which represents a downstream neural population that receives input from the top processing layer and performs a specific computation, such as recognition of a specific stimulus or classification of stimuli. For concreteness, we will use a layer of one or more readout binary neurons that perform binary classifications on the inputs. For simplicity, all neurons in the network are binary units, i.e., the activity level of each neuron is either 0 (silent) or 1 (firing). We denote $S_l^i \in \{0, 1\}$, the activity of the $i \in \{1, \ldots, N_l\}$ neuron in the $l = \{1, \ldots, L\}$ layer; $N_l$ denotes the size of the layer. The level of sparsity of the neural code, i.e. the fraction $f$ of active neurons for each stimulus, is set by tuning the threshold $T_l$ of the neurons in each layer (see below). For simplicity we will assume all neurons (except for the readout) have the same sparsity, $f$ .

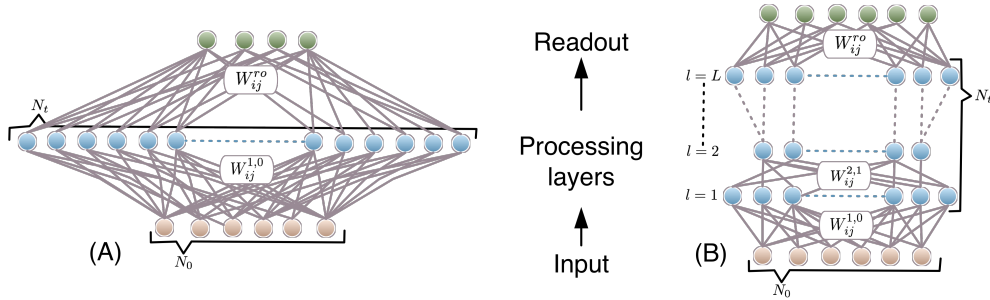

Figure 1: **Schematics of the network**. The network receives input from $N_0$ neurons and then projects them onto an intermediate layer composed of $N_t$ processing neurons. The neurons can be arranged in a single (A) or multiple (B) layers. The readout layer receives input from the last processing layer.

**Input**  The input to the network is organized as clusters around $P$ activity patterns. At it center, each cluster has a prototypical representation of an underlying specific stimulus, denoted as $\bar{S}_{0,\mu}^i$, where $i = 1, ..., N_0$ , denotes the index of the neuron in the input layer $l = 0$, and the index $\mu = 1, ..., P$, denotes the pattern number. The probability of an input neuron to be firing is denoted by $f_0$. Other members of the clusters are noisy versions of the central pattern, representing natural variations in the stimulus representation due to changes in physical features in the world, input noise, or neural noise. We model the noise as iid Bernoulli distribution. Each noisy input $S_{0,\nu}^i$ from the $\nu$th cluster, equals $\bar{S}_{0,\nu}^i$ $(-\bar{S}_{0,\nu}^i)$ with probability $(1 + m_0)/2$, $((1 - m_0)/2)$ respectively. Thus, the average overlap of the noisy inputs with the central pattern, say $\mu = 1$ is

$$m_0 = \frac{1}{N_0 f(1-f)} \left\langle \sum_{i=1}^{N_0} \left(S_0^i - f\right) \left(\bar{S}_{0,1}^i - f\right) \right\rangle, \tag{1}$$

ranging from $m_0 = 1$ denoting the noiseless limit, to $m_0 = 0$ where the inputs are uncorrelated with the centers. Topologically, the inputs are organized into clusters with radius $1 - m_0$.

**Update rule**   The state $S_l^i$ of the $i$-th neuron in the $l > 0$ layer is determined by thresholding the weighted sum of the activities in the antecedent layer:

$$S_l^i = \Theta \left( h_l^i - T_l \right). \tag{2}$$

Here $\Theta$ is the step function and the field $h_l^i$ represent the synaptic input to the neuron

$$h_l^i = \sum_{j=1}^{N_{l-1}} W_{l,l-1}^{ij} \left( S_{l-1}^j - f \right). \tag{3}$$

where the sparsity $f$ is the mean activity level of the preceding layer (set by thresholding, Eq. (2)).

**Synaptic matrix**   A key question is how the connectivity matrix $W_{l,l-1}^{ij}$ is chosen. Here we construct the weight matrix by first allocating for each layer $l$, a set of $P$ random templates $\xi_{l,\mu} \in \{0,1\}^N$ (with mean activity $f$), which are to serve as the representations of the $P$ stimulus clusters in the layer. Next, $W$ has to be trained to ensure that the response, $\bar{S}_{l,\mu}$, of the layer $l$ to a noiseless inputs, $\bar{S}_{0,\mu}$, equals $\xi_{l,\mu}$ . Here we use an explicit recipe to enforce these relations, namely the *pseudo-inverse* (PI) model [10, 8, 6], given by

$$W_{l,l-1}^{ij} = \frac{1}{N_{l-1}f(1-f)} \sum_{\mu,\nu=1}^{P} \left( \xi_{l,\nu}^i - f \right) \left[ C^{l-1} \right]_{\mu\nu}^{-1} \left( \xi_{l-1,\mu}^j - f \right), \tag{4}$$

where

$$C_{\mu\nu}^l = \frac{1}{N_l f(1-f)} \sum_{i=1}^{N_l} \left( \xi_{l,\mu}^i - f \right) \left( \xi_{l,\nu}^i - f \right) \tag{5}$$

is the correlation matrix of the random templates in the $l$th layer. For completeness we also denote $\xi_{0,\mu} = \bar{S}_{0,\mu}$. This learning rule guarantees that for noiseless inputs, i.e., $S_0 = \xi_{0,\mu}$, the states of all the layers are $S_{l,\mu} = \xi_{l,\mu}$. This will in turn allow for a perfect readout performance if noise is zero. The capacity of this system is limited by the rank of $C^l$ so we require $P < N_l$ [8].

A similar model of clustered inputs fed into a single processing layer has been studied in [1] using a simpler, Hebbian projection weights.

## 3   Mean Field Equations for the Signal Propagation

To study the dynamics of the signal along the network layers, we assume that the input to the network is a noisy version of one of the clusters, say, cluster $\mu = 1$. In the notation above, the input is a state $\{S_0^i\}$ with an overlap $m_0$ with the pattern $\xi_{0,1}$. Information about the cluster identity of the input is represented in subsequent layers through the overlap of the propagated state with the representation of the same cluster in each layer; in our case, the overlap between the response of the layer $l$, $S_l$, and $\xi_{l,1}$ , defined similarly to Eq. (1), as:

$$m_l = \frac{1}{N_l f(1-f)} \left\langle \sum_{i=1}^{N_l} \left( S_l^i - f \right) \left( \xi_{l,1}^i - f \right) \right\rangle. \tag{6}$$

In each layer the *load* is defined as

$$\alpha_l = \frac{P}{N_l}. \tag{7}$$

Using analytical mean field techniques (detailed in the supplementary material), exact in the limit of large $N$, we find a recursive equation for the overlaps of different layers. In this limit the fields and the fluctuations of the fields $\delta h_l^i$, assume Gaussian statistics as the realizations of the noisy input vary. The overlaps are evaluated by thresholding these variables, given by

($l \geq 2$)

$$m_{l+1} = H\left[\frac{T_{l+1} - (1-f)m_l}{\sqrt{\Delta_{l+1} + Q_{l+1}}}\right] - H\left[\frac{T_{l+1} + fm_l}{\sqrt{\Delta_{l+1} + Q_{l+1}}}\right], \tag{8}$$

where $H(x) = (2\pi)^{-1/2} \int_x^\infty dx \exp(-x^2/2)$. The threshold $T_l$ is set for each layer by solving

$$f = fH\left[\frac{T_{l+1} - (1-f)m_l}{\sqrt{\Delta_{l+1} + Q_{l+1}}}\right] + (1-f)H\left[\frac{T_{l+1} + fm_l}{\sqrt{\Delta_{l+1} + Q_{l+1}}}\right]. \tag{9}$$

The factor $\Delta_{l+1} + Q_{l+1}$ is the variance of the fields $\left\langle \left(\delta h_{l+1}^i\right)^2 \right\rangle$ which has two contributions. The first is due to the *variance* in the noisy responses of the previous layers, yielding

$$\Delta_{l+1} = f(1-f)\frac{\alpha_l}{1-\alpha_l}\left(1 - m_l^2\right). \tag{10}$$

The second contribution comes from the spatial correlations between noisy responses of the previous layers, yielding

$$Q_{l+1} = \frac{1 - 2\alpha_l}{2\pi(1-\alpha_l)}\left(f\exp\left[-\frac{(T_l - (1-f)m_{l-1})^2}{2(\Delta_l + Q_l)}\right] + (1-f)\exp\left[-\frac{(T_l + fm_{l-1})^2}{2(\Delta_l + Q_l)}\right]\right)^2. \tag{11}$$

Note that despite the fact that the noise in the different nodes of the input layer is uncorrelated, as the signals propagate through the network, correlations between the noisy responses of different neurons in the same layer emerge. These correlations depend on the particular realization of the random templates, and will average to zero upon averaging over the templates. Nevertheless, they contribute a non-random contribution to the total variance of the fields at each layer. Interestingly, for $\alpha_l > 1/2$ this term becomes negative, and reduces the overall variance of the fields.

The above recursion equations hold for $l \geq 2$. The initial conditions for this layer is $Q_1 = 0$ and $m_1$, $\Delta_1$ given by:

(Layer 1)

$$m_1 = H\left[\frac{T_1 - (1-f)m_0}{\sqrt{\Delta_1}}\right] - H\left[\frac{T_1 + fm_0}{\sqrt{\Delta_1}}\right], \tag{12}$$

$$f = fH\left[\frac{T_1 - (1-f)m_0}{\sqrt{\Delta_1}}\right] + (1-f)H\left[\frac{T_1 + fm_1}{\sqrt{\Delta_1}}\right], \tag{13}$$

and

$$\Delta_1 = f(1-f)\frac{\alpha_0}{1-\alpha_0}\left(1 - m_0^2\right). \tag{14}$$

where $\alpha_0 = P/N_0$.

Finally, we note that a previous analysis of the feedforward *PI* model (in the dense case, $f = 0.5$) reported results [6] neglected the contribution $Q_l$ of the induced correlations to the field variance. Indeed, their approximate equations fail to correctly describe the behavior of the system. As we will show, our recursion relations fully accounts for the behavior of the network in the limit of large $N$.

**Infinitely deep homogeneous network**   The above equations, eq (8)-(11) describe the dynamics of the average overlap of the network states and the variance in the inputs to the neurons in each layer. This dynamics depends on the sizes (and sparsity) of the different processing layers. Although the above equations are general, from now on, we will assume *homogeneous architecture* in which $N_l = N = N_t/L$ (all with the same sparsity). To find the behavior of the signals as they propagate along this *infinitely deep* homogenous network ($l \to \infty$) we look for the fixed points of the recursion equation.

Solution of the equations reveals three fixed points of the trajectories. Two of them are stable fixed points, one at $m = 0$ and the other at $m = 1$. The third is an unstable fixed point at some intermediate

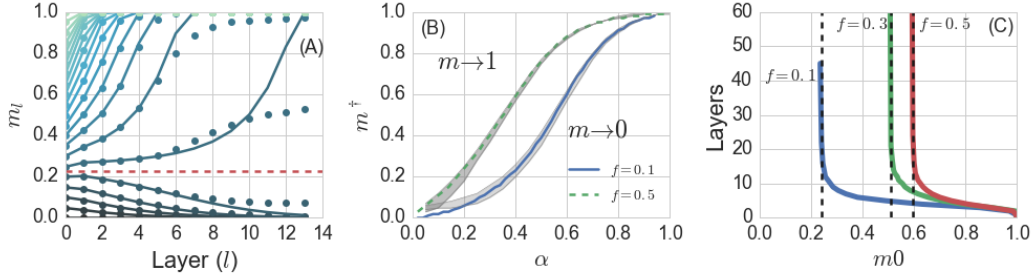

Figure 2: **Overlap dynamics.** (A) Trajectory of overlaps across layers from eq (8)-(11) (solid lines) and simulations (circles). Dashed red line show the predicted separatrix $m^{\dagger}$. The deviation from the theoretical prediction near the separatrix are due to final size effects of the simulations ($\alpha = 0.4$, $f = 0.1$). (B) Basin of attraction for two values of $f$ as a function of $\alpha$. Line show theoretical prediction and shaded area simulations. (C) Convergence time (number of layers) of the $m = 1$ attractor. Near the unstable fixed point (dashed vertical lines) convergence time diverges and rapidly decreases for larger initial conditions, $m_0 > m^{\dagger}$.

value $m^{\dagger}$. Initial conditions with overlaps obeying $m_0 > m^{\dagger}$ converge to 1, implying complete suppression of the input noise, while those with $m_0 < m^{\dagger}$ lose all overlap with the central pattern [figure 2.A], which depicts the values of the overlaps for different initial conditions. As expected, the curves (analytical results derived by numerically iterating the above mean field equations) terminate either at $m_l = 1$ or $m_l = 0$ for large $l$ . The same holds for the numerical simulations (dots) except for a few intermediate values of initial conditions that converge to an intermediate asymptotic values of overlaps. These intermediate fixed points are 'finite size effects'. As the system size ($N_t$ and correspondingly $N$) increases, the range of initial conditions that converge to intermediate fixed points shrinks to zero. In general increasing the sparsity of the representations (i.e., reducing $f$ ) improves the performance of the network. As seen in [figure 2.B] the basin of attraction of the noiseless fixed point increases as $f$ decreases.

**Convergence time** In general, the overlaps approach the noiseless state relatively fast, i.e., within $5 - 10$ layers. This holds for initial conditions well within the basin of attraction of this fixed point. If the initial condition is close to the boundary of the basin, i.e., $m_0 \approx m^{\dagger}$, convergence is slow. In this case, the convergence time diverges as $m_0 \to m^{\dagger}$ from above [figure 2.C].

## 4 Optimal Architecture

We evaluate the performances of the network by the ability of readout neurons to correctly perform randomly chosen binary linear classifications of the clusters. For concreteness we consider the performance of a single readout neuron to perform a binary classification where for each central pattern, the desired label is $\xi_{ro,\mu} = 0, 1$. The readout weights, projecting from the last processing layer into the readout [figure 1] are assumed to be learned to perform the correct classification by a *pseudo-inverse* rule, similar to the design of the processing weight matrices. The readout weight matrix is given by

$$W_{ro}^{j} = \frac{1}{N f_{ro}(1 - f_{ro})} \sum_{\mu,\nu=1}^{P} (\xi_{ro,\mu} - f_{ro}) \left[C^{L}\right]_{\mu\nu}^{-1} \left(\xi_{L,\mu}^{j} - f\right). \tag{15}$$

We assume the readout labels are iid Bernoulli variables with zero bias ($f_{ro} = 0.5$), though a bias can be easily incorporated. The error of the readout is the probability of the neuron being in the opposite state than the labels.

$$\epsilon = \frac{1 - m_{ro}}{2}, \tag{16}$$

where $m_{ro}$ is the average overlap of the readout layer, and can be calculated using the recursion equations (8)-(11). However, Since generally $f \neq f_{ro}$, the activity factor need to be replaced in the

proper positions in the equations. For correctness, we bring the exact form of the readout equation in the supplementary material.

## 4.1  Single infinite layer

In the following we explore the utility of deep architectures in performing the above tasks. Before assessing quantitatively different architectures, we present a simple comparison between a single infinitely wide layer and a deep network with a small number of finite-width layers.

An important result of our theory is that for a model with a single processing layer with finite $f$, the overlap $m_1$ and hence the classification error do not vanish even for a layer with infinite number of neurons. This holds for all levels of input noise, i.e., as long as $m_0 < 1$. This can be seen by setting $\alpha = 0$ in equations (8)-(11) for $L = 2$ . Note that although the variance contribution to the noise in the field, $\Delta_{ro}$ vanishes, the contribution from the correlations, $Q_1$, remains finite and is responsible for the fact that $m_{ro} < 1$ and $\epsilon > 0$ [1]. In contrast, in a deep network, if the initial overlap is within the basin of attraction the $m = 1$ solution, the overlap quickly approach $m = 1$ [figure (2).C]. This suggests that a deep architecture will generally perform better than a single layer, as can be seen in the example in figure 3.A.

**Mean error**    The readout error depends on the level of the initial noise (i.e., the value of $m_0$). Here we introduce a global measure of performance, $E$ , defined as the *readout error averaged over the initial overlaps*,

$$E = \int_0^1 dm_0 \rho\left(m_0\right) \epsilon\left(m_0\right),  \tag{17}$$

where the $\rho(m_0)$ is the distribution of cluster sizes. For simplicity we use here a uniform distribution $\rho = 1$. The mean error is a function of the parameters of the network, namely the sparsity $f$ , the input and total loads $\alpha_0 = P/N_0$, $\alpha_t = P/N_t$ respectively, and the number of layers $L$, which describes the layout of the network. We are now ready to compare the performance of different architectures.

## 4.2  Limited resources

In any real setting, the resources of the network are limited. This may be due to finite number of available neurons or a limit on the computational power. To evaluate the optimal architecture under constraints of a fixed total number of neurons, we assume that the total number of neurons is fixed to $N_t = \kappa N_0$, where $N_0$ is the size of the input layer. As in the analysis above, we consider for simplicity alternative uniform architectures in which all processing layers are of equal size $N = N_t/L$ . The performance as a function of the number of layers is shown in figure 3.B which depicts the mean error against the number of processing layers $L$ for several values of the expansion factor$\kappa$. These curves show that the error has a minimum at a finite depth

$$L_{opt} = \arg\min_L E(L).  \tag{18}$$

The reason for this is that for shallower networks, the overlaps have not been iterated sufficient number of times and hence remain further from the noiseless fixed point. On the other hand, deeper networks will have an increased load at each layer, since

$$\alpha = \frac{P}{\kappa N_0}L,  \tag{19}$$

thereby reducing the noise suppression of each layer. As seen in the figure, increasing the total number of neurons, yields a lower mean error $E_{opt}$, and increases the the optimal depth on the network. Note however, that for large $\kappa$ , the mean error rises slowly for $L$ larger than its optimal value; this is is because the error changes very slowly with $\alpha$ for small $\alpha$. and remains close to its $\alpha = 0$ value. Thus, increasing the depth moderately above $L_{opt}$ may not harm significantly the performance. Ultimately, if $L$ increases to the order of $\kappa N/P$ , the load in each processing layer $\alpha$ approaches 1, and the performance deteriorates drastically. Other considerations, such as time required for computation may favor shallower architectures, and in practice will limit the utility of architectures deeper than $L_{opt}$.

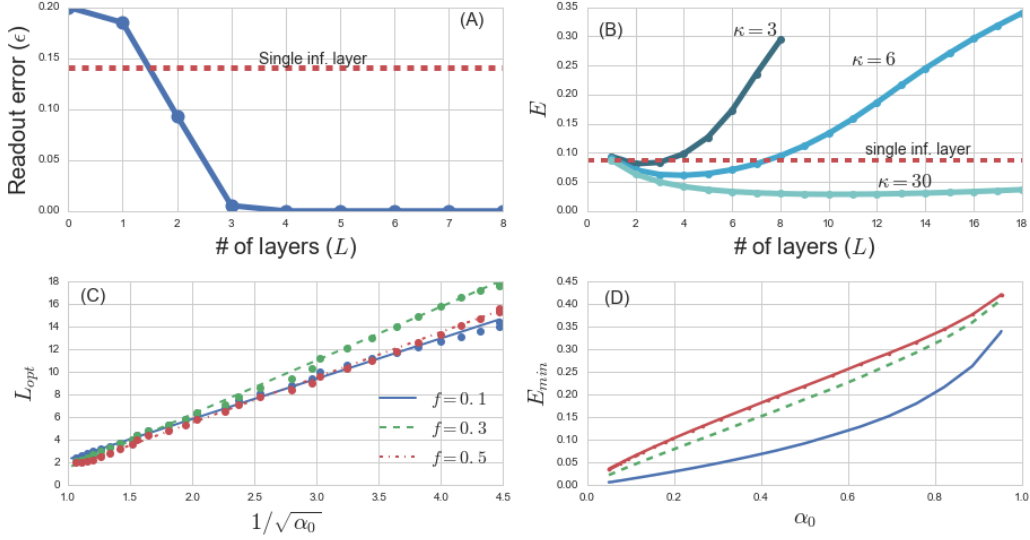

Figure 3: **Optimal layout**. (A) Comparing readout error produced by the same initial condition ($m_0 = 0.6$) of a single, infinitely-wide processing layer to that of a deep architecture with $\alpha = 0.2$. For both networks $\alpha_0 = 0.7$, $f = 0.15$ and $m_0 = 0.6$. (B) Mean error as a function of the number of the processing layers for three values of expansion factor $\kappa = N_t/N_0$. Dashed line shows the error of a single infinite layer. (C) Optimal number of layers as a function of the inverse of the input load ($\alpha_0 \propto P$), for different values of sparsity. Lines show linear regression on the data points. (D) minimal error as a function of the input load (number of stored templates). Same color code as (C).

**The effect of load on the optimal architecture**   If the overall number of neurons in the network is fixed, then the optimal layout $L_{opt}$ is a function of the size of the dataset, i.e, $P$. For large $P$, the optimal network becomes shallow. This is because that when the load is high, resources are better allocated to constrain $\alpha$ as much as possible, due to the high readout error when $\alpha$ is close to 1, figures C and D . As shown in [figure 3.D], $L_{opt}$ increases with decreasing the load, scaling as

$$L_{opt} \propto P^{-1/2}. \tag{20}$$

This implies that the width $N_{opt}$ scales as

$$N_{opt} \propto P^{1/2}. \tag{21}$$

## 4.3   Autoencoder example

The model above assumes inputs in the form of random patterns ($\xi_{0,\mu}$) corrupted by noise. Here we illustrate that the qualitative behavior of the network for inputs generated by handwritten digits (MNIST dataset) with random corruptions. To visualize the suppression of noise by the deep pseudo-inverse network, we train the network with autoencoder readout layer, namely use a readout layer of size $N_0$ and readout labels equal the original noiseless images, $\xi_{ro,\mu} = \xi_{0,\mu}$. The readout weights are Pseudo-inverse weights with output labels identical to the input patterns, and following eq. (15). [**?** 2]. A perfect overlap at the readout layer implies perfect reconstruction of the original noiseless pattern.

In figure 4, two networks were trained as autoencoders on a set of templates composed of 3-digit numbers (See experimental procedures in the supplementary material). Both networks have the same number of neurons. In the first, all processing neurons are placed in a single wide layer, while in the other neurons were divided into 10 equally-sized layers. As the theory predicts, the deep structure is able to reproduce the original templates for a wide range of initial noise, while the single layer typically reduces the noise but fails to reproduce the original image.

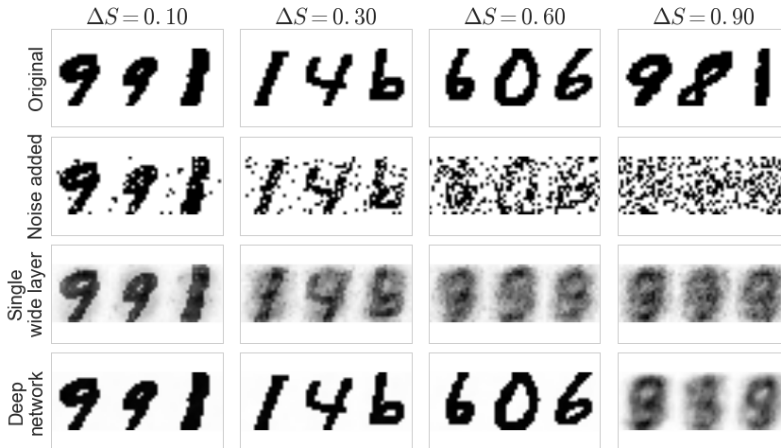

Figure 4: **Visual example of the difference between a single processing layer and a deep structure**. Input data was prepared using the MNIST handwritten digit database. Example of the templates are shown on the top row. Two different networks were trained to autoencode the inputs, one with all the processing neurons in a single layer (figure 1.A) and one in which the neurons were divided equally between 10 layers (figure 1.B) (See experimental procedures in the supplementary material for details). A noisy version of the templates were introduced to the two networks and the outputs are presented on the third and fourth rows, for different level of initial noise (columns).

## 5   Summary and Final Remarks

Our paper aims at gaining a better understanding of the functionality of deep networks. Whereas the operation of the bottom (low level processing of the signals) and the top (fully supervised) stages are well understood, an understanding of the rationale of multiple intermediate stages and the tradeoffs between competing architectures is lacking. The model we study is simplified both in the task, suppressing noise, and its learning rule (pseudo-inverse). With respect to the first, we believe that changing the noise model to the more realistic variability inherent in objects will exhibit the same qualitative behaviors. With respect to the learning rule, the pseudo-inverse is close to SVM rule in the regime we work, so we believe that is a good tradeoff between realism and tractability. Thus, although the unavoidable simplicity of our model, we believe its analysis yields important insights which will likely carry over to the more realistic domains of deep networks studied in ML and neuroscience.

**Effects of sparseness**   Our results show that the performance of the network is improved as the sparsity of the representation increases. In the extreme case of $f \to 0$, perfect suppression of noise occurs already after a single processing layer. Cortical sensory representations exhibit only moderate sparsity levels, $f \approx 0.1$. Computational considerations of robustness to 'representational noise' at each layer will also limit the value of $f$. Thus, deep architectures may be necessary for good performance at realistic moderate levels of sparsity (or for dense representations).

**Infinitely wide shallow architectures:**   A central result of our model is that a finite deep network may perform better than a network with a single processing layer of infinite width. An infinitely wide shallow network has been studied in the past (e.g., [4]). In principle, an infinitely wide network, even with random projection weights, may serve as a universal approximate, allowing for yielding readout performance as good as or superior to any finite deep network. This however requires a complex training of the readout weights. Our relatively simple readout weights are incapable of extracting this information from the infinite, shallow architecture. Similar behavior is seen with simpler readout weights, the Hebbian weights as well as with more complex readout generated by training the readout weights using SVMs with noiseless patterns or noisy inputs [1]. Thus, our results hold qualitatively for a broad range of plausible readout learning algorithms (such as Hebb, PI, SVM) but not for arbitrarily complex search that finds the optimal readout weights.

## Acknowledgements

This work was partially supported by IARPA (contract #D16PC00002), Gatsby Charitable Foundation, and Simons Foundation SCGB grant.

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
