[Supplementary Material]

# Optimal Architectures in a Solvable Model of Deep Networks: Supplementary Material

## 1 Experimental procedures

### 1.1 Determining the threshold

In each layer $l$, the threshold $T_l$ is calculated so the desired sparsity is maintained [eq (9) in the main text]. A fixed threshold implies a mean activity $f$ when averaged over many realizations of noise. For practical reasons in simulations, we strictly enforce sparsity by setting the activity of the $fN$ neurons with the highest local fields $h^i$ to 1, while silencing the remaining $(1 - f)N$ neurons. For large systems, where the fluctuations of the mean activity around $f$ are negligible, this approximation is both valid and highly efficient.

### 1.2 MNIST data set

For a visual example of the processing by de-nosing network which we study, we have created a set of binary images, based on the well known MNIST [1] dataset for handwritten digits. The model requires a large set of uncorrelated binary inputs. For this end we have crated a set of 1000 unique 3-digit numbers, where each image is composed of three digits taken from the MNIST dataset. Each image is then thresholded to get a binary $\{0, 1\}$ values for all the pixels. The result is a rectangular 1200 binary pixels image.

We trained network of various sizes and layouts to autoencode [2] a subset of size $P$ of the dataset. Varying $P$ allows to change the input load $\alpha_0 = P/1200$. For each trained network, noisy samples of the input were tested. A noisy input was created by adding Bernuli noise in order to obtain the desired initial overlap with the original pattern, $m_0$. The output of the network was of equal size to the input layer, $N_{ro} = 1200$ and was trained to re-encode the input images from the final processing layer.

In figure 4 in the main text, we show typical example of the results. In that example we have trained two networks with $P = 800$ ($\alpha_0 = 0.66$) input images. Each network has $N_t = 72000$ intermediate neurons giving expansion factor $\kappa = 60$. In one network all neurons were organized in a single layer layout, while in the other 'deep' network, they were stacked in $L = 10$ equally sized

layers. The output presented in the figure are the thresholded fields on the readout neurons. The activity level of the readout layer $f_{ro}$ was chosen to match the input pattern, to allow autoencoding of the images.

It is important to note that this network does not preform any classification. The MNIST dataset is used here only as an accessible bank of binary images used to illustrate the performances of the networks in terms of noise reduction.

## 2   Mean field derivation of the recursion equations

Here we provide the derivation of the recursive equations using mean field derivation. For convenience of notation we denote zero averaged variables $x_{\mu,l}^i = \xi_{\mu,l}^i - f$ and $s_l^i = S_l^i - f$, where $f$ is the sparsity level of all layers (except possibly the input layer). The templates $\xi_{\mu,l}^i = \{0,1\}$ are binary variables associated with the random sparse pattern allocated at layer $l$ for for the $\mu$-th cluster. Similarly, $S_l^i$ is the activation of layer $l$ neurons resulted by propagating along the layers a given input. We consider the case where the input is a noisy version of one of the input cluster centers, say cluster $1$. Thus, $S_l^i$ $=\xi_{1,0}^i$ with probability $(1 + m_0)/2$ and $-\xi_{1,0}^i$ with probability $(1 - m_0)/2$. We are interested in the statistics of the fields at all layers, induced by this input. In the large $N$ limit the field induced on the neuron $i$ of the $l-th$ layer obeys a gaussian statistics. characterized by a mean $\langle h_l^i \rangle$ and variance $\sigma_l^2$.

The overlap with the central pattern in each layer is given by (eq. (6) in the main text)

$$m_l = \frac{1}{Nf(1-f)} \sum_{i=1}^N \langle x_{1,l}^i s_l^i \rangle_l = \frac{1}{Nf(1-f)} \sum_{i=1}^N \langle x_{1,l}^i \left( \Theta \left[ h_l^i - T_l \right] - f \right) \rangle_l. \quad (1)$$

Here the angular brackets denote explicit average over the random templates the layer $l$, while averaging on previous layers is implicit in the calculation of the fields $h_l$ over the statistics of the precedent layers. To preform the average, we note that the random templates $\xi_{\mu,l}$ obey the sparseness requirements, namely that only a fraction $f$ of the bits are 1, then average over $x_{\mu,l}^i$ can be broken into two contributions: a fraction $f$ with an average of $(1 - f)$ and a fraction $(1 - f)$ with an average of $(-f)$. Taking this statistics into account, we can write eq (1) as

$$m_l = \frac{1}{N} \sum_{i=1}^N \langle \Theta \left[ h_l^+ - T_l \right] - \Theta \left[ h_l^- - T_l \right] \rangle_h \quad (2)$$

where $h_l^{i+}$ $(h_l^{i-})$ implies the mean input to neuron $i$ in the case that $\xi_{1,l}^i = 0$ $(\xi_{1,l}^i = 1)$. Performing the gaussian integral over the fields results in

$$m_l = H\left[\frac{T_l - \langle h_l^+ \rangle}{\sqrt{\sigma_l^2}}\right] - H\left[\frac{T_{l+1} - \langle h_l^- \rangle}{\sqrt{\sigma_l^2}}\right], \tag{3}$$

where $H(x) = \int_x^\infty dx \exp(-x^2/2)$. Using the same arguments, we can write an equation for the mean activity level in each layer, which by construction is equal to $f$,

$$f = f\left[\frac{T_l - \langle h_l^+ \rangle}{\sqrt{\sigma_l^2}}\right] + (1-f)H\left[\frac{T_l - \langle h_l^- \rangle}{\sqrt{\sigma_l^2}}\right]. \tag{4}$$

By solving (4), the correct threshold $T_l$ for each layer, which results in activity $f$ is found.

The theory provides recursion relations for the means and variances of the field

$$h_l^i = \sum_j W_{l,l-1}^{ij} s_j^{l-1} \tag{5}$$

where the weight matrix is defined as

$$W_{l,l-1} = \frac{1}{Nf(1-f)} X^l [C^{l-1}] X^{(l-1)T}. \tag{6}$$

Here we denote $X^l$ the $N \times P$ matrix of than random patterns $\{x_{\mu,l}^i\}$, and $C^l = \frac{1}{Nf(1-f)} X^{lT} X^l$, is a $P \times P$ matrix of their correlations. In eq (3) and (4), $\sigma_l^2 \equiv \langle \delta h_l^2 \rangle$ is the variance of the fluctuations in the field.

We can write zero averaged activity $s_j^{l-1}$ as a mean and fluctuations

$$s_j^{l-1} = m_{l-1} x_{1,l-1}^j + \delta s_j^{l-1}, \tag{7}$$

where the mean of $\delta s_j^{l-1}$ is zero. By accounting for the binary sparse statistics of $s_j^{l-1}$ we havee

$$\langle (\delta s_j^{l-1})^2 \rangle = f(1-f)(1 - m_{l-1}^2). \tag{8}$$

Thus,

$$\langle h_l^i \rangle = m_{l-1} \sum_j W_{l,l-1}^{ij} x_{1,l-1}^j = m_{l-1} x_{1,l-1}^i \tag{9}$$

by virtue of the fact that the summation over the input index $j$ yields $\sum_j x_{\nu,l-1}^j x_{1,l-1}^j = Nf(1-f)C_{\nu,1}^{l-1}$. It follows that

$$\langle h_l^+ \rangle = (1-f)m_{l-1}, \ \langle h_l^- \rangle = -fm_{l-1}. \tag{10}$$

## 2.1 Fluctuations

The variance of the fields is given by

$$
\sigma_l^2 \equiv \left\langle \left(\delta h_l^i\right)^2 \right\rangle =
$$
$$
\frac{1}{N^3 f^2 (1-f)^2} Tr_N \left\langle X^l \left[C^{l-1}\right]^{-1} X^{(l-1)T} \langle \delta s^{l-1} \delta s^{l-1T} \rangle X^{Tl} [C^{l-1}]^{-1} X^{-(l-1)} \right\rangle. \tag{11}
$$

The inner averaging,

$$
\langle \delta s^{l-1} \delta s^{l-1T} \rangle, \tag{12}
$$

is over the noise injected by the noisy inputs; the external average is over patterns (since this quantity is self-averaging).

First, it is straightforward to average over the patterns in the $l$-th layer, $X^l$, yielding,

$$
\sigma_l^2 = \frac{1}{N^2 f(1-f)} Tr_P \left\langle \left[C^{l-1}\right]^{-2} X^{(l-1)T} \langle \delta s^{l-1} \delta s^{l-1T} \rangle X^{(l-1)} \right\rangle. \tag{13}
$$

The challenge is to compute the average over the signal (12). We separate this average into two contributions,

$$
\langle \delta s^{l-1} \delta s^{l-1T} \rangle = f(1-f)(1-m_{l-1}^2) + q^{l-1}. \tag{14}
$$

The first term is due to variance in the distribution of overlaps across patterns. Its contribution to the total variance of the fields is given by

$$
\Delta_l = \frac{f(1-f)(1-m_{l-1}^2)}{N^2 f(1-f)} Tr_P \langle \left[C^{l-1}\right]^{-2} X^{(l-1)T} X^{(l-1)} \rangle
$$
$$
= \frac{f(1-f)(1-m_{l-1}^2)}{N} Tr_P \langle \left[C^{l-1}\right]^{-1} \rangle
$$
$$
= \frac{\alpha f(1-f)(1-m_{l-1}^2)}{1-\alpha}. \tag{15}
$$

The second term, $q^{l-1}$, contains all the non-diagonal terms in the correlation matrix, and expresses cross-correlations in the noise.

## 2.2 Noise correlations

The expression in (15) would be the only contribution if there were no correlations in the noise. Indeed in the input layer the noise is assumed uncorrelated.

However as a noisy pattern (i.e., a pattern that does not perfectly overlap the central template) propagate through the layers, spatial correlations form, which cannot be neglected. The values in the matrix $q^{l-1}$ in (14) are fluctuating with the patterns $X$, and their mean, once the quenched average over the patterns is taken, vanishes. Nevertheless, their contribution to (11) is non-zero, and cannot be neglected[1]. To compute this contribution, we examine individual terms for a given $j$ and $j'$,

$$Z^{l-1} = \frac{1}{Nf(1-f)} Tr_P \left[ C^{l-1} \right]^{-2} X^{(l-1),jT} X^{(l-1),j'}$$

$$\equiv \alpha u^{(l-1)T} \left[ C^{l-1} \right]^{-2} v^{l-1} \quad (16)$$

where

$$u_\mu^{l-1} = x_\mu^{(l-1),j} / \sqrt{Pf(1-f)}, \quad (17)$$

$$v_\mu^{l-1} = x_\mu^{(l-1),j'} / \sqrt{Pf(1-f)} \quad (18)$$

are two $P$-dimensional vectors, normalized so that $u^T u = v^T v = 1$. Furthermore, we have

$$u^T v \sim \mathcal{O} \left( \frac{1}{\sqrt{\alpha N}} \right). \quad (19)$$

Below, we will use this property to derive the results for large $N$. We can write the pattern correlation matrix $C$ itself as

$$C = C_0 + \alpha \left( uu^T + vv^T \right) \quad (20)$$

where $C_0$ is the contributions from all sites other than $j$ and $j'$.

In here and the following we temporally suppress the layer index superscript without ambiguity, and remember that $X$ and $C$ are defined independently for each layer. The inverse of eq (20) can be written as

$$C^{-2} = [I + \delta]^{-2} C_0^{-2}, \quad (21)$$

where $I$ is the $P \times P$ identity matrix, and

$$\delta = \alpha C_0^{-1} \left( uu^T + vv^T \right). \quad (22)$$

Since $C_0$ is a random matrix, independent of the values in the sites $j$ and $j'$, we allow averaging over it, resulting in [4]

$$\langle C_0^{-1} \rangle = \frac{1}{1-\alpha} I \quad (23)$$

$$\langle C_0^{-2} \rangle = \frac{1}{(1-\alpha)^3} I. \quad (24)$$

Eq. (21) can be represented as a power series in the matrix $\delta$ as

$$C^{-2} = \frac{1}{(1-\alpha)^3} \left[ I - 2\delta + 3\delta^2 - 4\delta^3 + \dots \right]. \tag{25}$$

Substituting into equation (16), we get

$$Z = \frac{1}{(1-\alpha)^3} u^T \left[ I - 2\delta + 3\delta^2 - 4\delta^3 + \dots \right] v^T. \tag{26}$$

Using the expression for $\delta$ from (22), and keeping only leading orders in $u^T v$, we get

$$Z = \frac{1}{(1-\alpha)^3} \left[ 1 - 2^2 \frac{\alpha}{1-\alpha} + 3^2 \left( \frac{\alpha}{1-\alpha} \right)^2 - 4^2 \left( \frac{\alpha}{1-\alpha} \right)^3 + \dots \right] u^T v$$

$$= \frac{1-2\alpha}{1-\alpha} u^T v. \tag{27}$$

The factor in the last equality of (27) results from the sum of an infinite power series in $\alpha/(1-\alpha)$. Finally, substituting this results in eq., (13) for the full variance

$$\sigma^2 = \Delta_l + Q_l \tag{28}$$

where

$$Q_l = \frac{(1-2\alpha)}{(1-\alpha)} \frac{1}{N^2 f(1-f)} \left\langle \sum_{\nu,j \neq j'} x_\nu^{(l-1),j} x_\nu^{(l-1),j'} \delta s_j^{l-1} \delta s_{j'}^{l-1} \right\rangle \tag{29}$$

Finally, noting that each $x_j^{l-1}$ are effectively Gaussian, we can write the average

$$\frac{1}{\sqrt{Nf(1-f)}} \left\langle \sum_{,j} x_\nu^{(l-1),j} \delta s_j^{l-1} \right\rangle = \sigma_{l-1} \frac{\partial}{\partial h_j^{l-1}} \langle \delta s_j^{l-1} \rangle$$

$$= \sqrt{\sigma_{l-1}} \frac{\partial}{\partial z} \langle \Theta \left( \langle h_{l-1} \rangle + z\sqrt{\sigma_{l-1}} - T_{l-1} \right) \tag{30}$$

yielding

$$Q_l = \frac{1-2\alpha}{(1-\alpha)} \left( f H' \left[ \frac{(T_{l-1} - (1-f)m_{l-1})}{\sigma_{l-1}} \right] + (1-f) H' \left[ \frac{(T_{l-1} + m_{l-1})}{\sigma_{l-1}} \right] \right)^2. \tag{31}$$

where,

$$H'(x) = -\frac{1}{\sqrt{2\pi}} \exp\left( -x^2/2 \right). \tag{32}$$

# 3    Readout weights

Since the readout has a different sparsity $f_{ro} \neq f$ , it is important to take notice on which sparsity factor appears where in the update equation. For correctness we bring here the exact expression for the readout weights

$$m_{ro} = H\left[\frac{T_{ro} - (1 - f_{ro})m_L}{\sqrt{\Delta_{ro} + Q_{ro}}}\right] - H\left[\frac{T_{ro} + f_{ro}m_L}{\sqrt{\Delta_{ro} + Q_{ro}}}\right], \tag{33}$$

$$f_{ro} = f_{ro}H\left[\frac{T_{ro} - (1 - f_{ro})m_l}{\sqrt{\Delta_{ro} + Q_{ro}}}\right] + (1 - f_{ro})H\left[\frac{T_{ro} + f_{ro}m_L}{\sqrt{\Delta_{l+1} + Q_{l+1}}}\right]. \tag{34}$$

$$\Delta_{ro} = f(1 - f)\frac{\alpha}{1 - \alpha}\left(1 - m_L^2\right). \tag{35}$$

and

$$Q_{ro} = \frac{1 - 2\alpha}{2\pi(1 - \alpha)} \times$$
$$\left(f\exp\left[-\frac{(T_L - (1 - f)m_{L-1})^2}{2(\Delta_L + Q_L)}\right] + (1 - f)\exp\left[-\frac{(T_L + fm_{L-1})^2}{2(\Delta_L + Q_L)}\right]\right)^2. \tag{36}$$

## Footnotes

[1] Contrary to the assumption of Meir and Domany, [3], where the contribution of the of those fluctuations was neglected.