[Reviews · NeurIPS 2016]

Reviewer 1

Summary

This paper describes a probabilistic "neural" model. It attempts to re-introduce a lot of existing concepts of RBMs and other unidrected graphical models. The model can generate digits like it's 2004. That is to say these experimental results are outdated.

Qualitative Assessment

Paper attempts to re-introduce a lot of existing concepts of RBMs and other unidrected graphical models. The model can generate digits like it's 2004. That is to say these experimental results are outdated.

Confidence in this Review

2-Confident (read it all; understood it all reasonably well)


Reviewer 2

Summary

This paper considers a multilayer network to reduce noise in the input pattern. Through mean field analysis, the authors got an analytical recursive relation of pattern overlaps. With it, the optimal layer number of the multilayer network could be obtained. The paper is an extension of the previous work [Baktash Babadi Haim Sompolinsky, 2014 Neuron], and apply the mean field method devised in that paper to reduce pattern noise by adding the layer number of the multilayer network. Except for noise reduction, are there any other merits for the multilayer organized network, e.g., feature extraction or pattern separation? There are a lot of typos and errors in the present version, where some are listed below: Line 56: activity y Line 67: Line 69: At it center Line 73: Eq.3, is it i=1, not j=1? Line 78: has to Line 145: figure 2D, where? Line 213: Figure 4: y labels

Qualitative Assessment

This paper mainly considers pattern robustness when propagating along network layers. The result shows that a network with multiple layers is beneficial for noise reduction. For the current version, this paper is quite similar to [Baktash Babadi Haim Sompolinsky, 2014 Neuron], especially for the technical analysis.

Confidence in this Review

2-Confident (read it all; understood it all reasonably well)


Reviewer 3

Summary

This paper uses simplified deep networks to study the propagation of signals through the networks. The authors find that performance trade-offs between depth and width depend on the number of templates that must be stored in the network. Furthermore, the authors find that performance increases with the sparsity of representation.

Qualitative Assessment

The paper was interesting despite not completely following all of it. The analysis of the trade-offs between depth and width are useful and interesting. It is unclear how the conclusions drawn would generalize to more realistic networks (non-binary activations trained with backprop), although it seems likely they would.

Confidence in this Review

1-Less confident (might not have understood significant parts)


Reviewer 4

Summary

The authors use a solvable model of deep networks to explore the dependence of the optimal architecture on system parameters like depth, sparsity and load.

Qualitative Assessment

It is a commendable effort to find solvable models that allow us to sharpen our intuition on how and when deep learning works. Yet, my concern is that the #patterns < #neurons regime the authors explore is not particularly relevant for deep learning. In particular: - The noise structure in the model is supposed to include noise due the natural variations in the world, but it is modeled only as binomial variability. The power of deep networks is to marginalize over natural variability in order to generalize and to extract the features of interest. Modeling this variability as independent noise prevents the network from doing any interesting generalizations across patterns. - The authors probe the network in a "low capacity" regime where #patterns < #neurons, which is not particularly relevant for most deep learning applications (or the brain) where typically #patterns >> #neurons. The surprising finding by the authors that shallower networks are better for large #patterns seems to be an artefact of the probed regime and noise.

Confidence in this Review

2-Confident (read it all; understood it all reasonably well)


Reviewer 5

Summary

This paper describes a particular instantiation of a deep neural network with solvable dynamics during feedforward propagation of an input. They show for such an architecture, an optimal depth (number of layers) can be computed and determine analytically situations where higher depth is beneficial.

Qualitative Assessment

While this work attempts to elucidate a theoretical understanding of deep networks, the model they analyze diverges substantially from models which have been successful. For example, the weights in their model are 'trained' analytically in a layer-wise manner (via pseudo-inverse rule). Furthermore, the authors presume the function of each layer is to preserve the fundamental clustering structure of the input while progressively removing noise. There is no citation or work provided showing this empirically occurs in deep networks. Unlike [1] which they cite as related theoretical work on deep learning, the do not connect their findings back to the models which are currently successful. Therefore, it is unclear how to connect their theoretical findings with the current field of deep learning. That aside, the analysis and results do have close connections to hebbian learning and more connections should be made with that literature. Otherwise, the current connection to modern deep networks made in this work is tenuous. [1] Andrew M Saxe, James L McClelland, and Surya Ganguli. Exact solutions to the nonlinear 274 dynamics of learning in deep linear neural networks. arXiv.org, December 2013.

Confidence in this Review

2-Confident (read it all; understood it all reasonably well)